# Combining Historical and Molecular Data to Study Nearly Extinct Native Italian Grey Partridge (*Perdix perdix*) at the Turn of the Twentieth Century

**DOI:** 10.3390/biology13090709

**Published:** 2024-09-10

**Authors:** Claudia Greco, Cristiano Tabarroni, Irene Pellegrino, Livia Lucentini, Leonardo Brustenga, Lorenza Sorbini, Nadia Mucci

**Affiliations:** 1Area for Conservation Genetics, BIO-CGE, Department Monitoring and Protection of the Environment and Conservation of Biodiversity, Italian National Institute for Environmental Protection and Research, ISPRA, Via Cà Fornacetta n°9, Ozzano dell’Emilia, 40064 Bologna, Italy; 2Department for Sustainable Development and Ecological Transition, DiSSTE, University of Eastern Piedmont, UNIUPO, Piazza Sant’Eusebio 5, 13100 Vercelli, Italy; 3Department of Chemistry, Biology and Biotechnology, University of Perugia, UNIPG, Via del Giochetto, 06123 Perugia, Italy; 4Department of Veterinary Medicine, University of Perugia, UNIPG, Via San Costanzo 4, 06126 Perugia, Italy; 5Library ISPRA, Italian National Institute for Environmental Protection and Research, Via Cà Fornacetta n°9, Ozzano dell’Emilia, 40064 Bologna, Italy

**Keywords:** grey partridge, Italy, haplotypes, museum samples, mtDNA

## Abstract

**Simple Summary:**

The grey partridge (*Perdix perdix* Linnaeus, 1758), is a polytypic species with seven recognized subspecies, including one subspecies (*P. p. italica* Hartert, 1917) that is endemic to the Italian peninsula. Until World War II, the species was widespread across Europe, but then severely declined due to anthropogenic causes, including hybridization. The contraction of the endemic population distribution range and the introduction of allochthonous genotypes that started at the beginning of the past century led to the impoverishment of the endemic gene pool. This is still one of the main threats to this species and is only partially managed through ex situ conservation programs. To understand the native genetic composition of Italian grey partridge, we performed genetic characterization of museum specimens, proving that museum samples are a reliable source of DNA for investigating the genetic structure of species or populations from the recent past. The data obtained highlight, along with a contraction in the distribution, a high presence of specimens coming from other European populations, suggesting an uncertain situation of the species in Italy, despite the conservation efforts made. Extensive historical bibliographic research allowed us to define time baselines and periods with different restocking pressures.

**Abstract:**

The grey partridge (*Perdix perdix* Linnaeus, 1758), is a polytypic species with seven recognized subspecies, including *P. p. italica* (Hartert, 1917), which is endemic to Italy. Until World War II, the species was widespread across Europe but severely declined due to anthropogenic causes, jeopardizing the Italian subspecies gene pool. Genetic characterization and haplotype identification were performed by analyzing the 5′-end of the mitochondrial control region (CR). A total of 15 haplotypes were detected, seven of which were present in the population before 1915. Among them, three haplotypes were never detected again in the individuals collected after 1915. Interestingly, eight of the 15 haplotypes detected in Italian museum samples belonged exclusively to individuals collected after 1915. The obtained data highlight a high presence of specimens originating from other European populations and, despite all the conservation efforts, suggest an uncertain situation of the subspecies in Italy. This research was strongly backed up by extensive bibliographic research on historical documents, allowing the identification of hundreds of restocking events all over Italy. This is an integral part of this research and has laid the foundations for identifying and circumscribing historical periods in which introductions from the rest of Europe had different pressures, aiming to define a baseline.

## 1. Introduction

Humans are the principal cause of biodiversity loss and species extinction [1]. From AD 1500, around 784 extinctions of mammals, birds, amphibians, fish, and invertebrates have been documented worldwide [2,3]. In 2016, Bull and Maron [3] showed that the main extinction events affected invertebrates (*n* = 359), then birds (*n* = 129), fishes (*n* = 81), mammals (*n* = 79), amphibians (*n* = 34), and reptiles (*n* = 21). Summary statistics of IUCN’s 2023 report [2], recorded an increased number of extinct species in several classes, specifying which of them disappeared from the wild (Extinct in the Wild, EW) but survived in ex situ collections (e.g., zoos, captivity, or seminatural conditions) [4]. As reported in the literature, ex situ conservation programs allowed to save many species from extinction [5]. For instance, in 2009, two captive populations of endemic Christmas Island lizard species, *Emoia atrocostata* (Lesson, 1830) and *Emoia nativitatis* (Boulenger, 1887), were established to save the species that vanished from the island in 2012 [5,6]. In 1925, 17 Greyson’s doves (*Zenaida graysoni* Lawrence, 1871) were collected from Socorro Island to form ex situ populations, which are still viable and distributed in captive breeding facilities across North America and Europe. Bolam et al. [7] showed that ex situ conservation was relevant and fundamental in preventing the extinction of several bird and mammal species. Evidence showed that although some of the species conserved in ex situ collections went completely extinct (e.g., *Megupsilon aporus* Miller and Walters, 1972) [5], others survived and were reintroduced into the wild, where they successfully formed new populations (e.g., *Bison bonasus* Linnaeus, 1758) [8].

One of the main threats leading *taxa* to extinction is the loss of the gene pool, driven by anthropogenic hybridization with other species or populations that are released, either accidentally or intentionally, into the wild for various purposes [9]. Releases usually lead to the admixture or hybridization of individuals belonging to autochthonous and allochthonous taxa, often causing species extinction or the loss of adaptative variants of native populations.

The grey partridge (*Perdix perdix* Linnaeus, 1758) is a medium-sized non-migratory bird with low dispersal abilities [10], living in open rural environments. For more information on the species, see Appendix A. It is classified as a polytypic species, with seven recognized subspecies, native to cold steppes and natural prairies but has adapted to live in agricultural environments and to higher temperatures [11]. Until World War II, the species was widespread across Europe [12] and then started to severely decline in 1950 [13], with a reduction of about 94% since the 1980s [10,14]. The main causes of decline are the limited insect availability due to pesticides, fundamental in the diet of chicks for the first months of life and during the breeding phase, overexploitation from hunters, and restocking activities with allochthonous individuals [13]. The species is categorized as Least Concern by the International Union for Conservation of Nature (IUCN) but is estimated to be declining (IUCN 2023; BirdLife International, 2004). The Italian subspecies, *P.p. italica* (Hartert, 1917) (Annex I of the Birds Directive 794/409/EEC; Annex III of Bern Convention on the Conservation of European Wildlife and Natural Habitats, 1979 (https://eur-lex.europa.eu/LexUriServ/LexUriServ.do?uri=CONSLEG:1979L0409:20070101:IT:PDF accessed on 1 August 2024)), was widespread over all peninsular elevations below 500 m asl, from alpine valleys to the Strait of Messina, and was never reported in Sicily, Sardinia and the smaller islands [15]. Strong hunting pressure and inadequate management strategies, with large-scale releases of birds that originated from captivity or that were imported from Europe (Denmark, France, Hungary, Slovenia, Romania, Poland), led to a strong reduction in the Italian populations, followed by an erosion of the genetic integrity of the endemic subspecies [16].

The Italian peninsula acted as a refugium for several species during the climatic oscillation of the Pleistocene [17] and several populations isolated, differentiated, or speciated in the peninsula: *Capreolus capreolus italicus* (Festa, 1925) [18], *Alectoris graeca whitakeri* (Schiebel, 1934) [19], *Lepus corsicanus* (de Winton, 1898) [20], *Passer italiae* (Vieillot, 1817) [21]. Climatic changes affected the distribution of cold-adapted species, which remained in patched and fragmented areas of the peninsula after temperatures increased: *Ichthyosaura alpestris* (Laurenti, 1768) [22], *Rupicapra pyrenaica ornata* (Neumann, 1899) [23]. Even if Italian grey partridge is a cold-adapted species, no information on how the climate changes have shaped its distribution and genetic pattern compared to other European populations has ever been provided.

In 2002, Liukkonen and colleagues analyzed a fragment of the mitochondrial control region in 227 European samples and provided the first evidence of phylogenetic relationships among European populations. Although the sampling effort was exhaustive to infer the population structure at the European level, Italy was poorly represented, with only five wild individuals from Northern Italy tested [24]. Notwithstanding the small and unrepresentative sample, two individuals from Italy shared a unique haplotype, thus suggesting an isolation of the Italian populations during the past climatic oscillations. However, due to the contraction of the endemic population and the introduction of allochthonous genotypes started at the beginning of the past century, the genetic pattern of the Italian grey partridge cannot be recorded, and no assessment regarding the effects of Pleistocene climatic oscillations on the Italian subspecies can be documented. The absence of scientific evidence on the native gene pool hampered any chance to verify this information.

The only valuable source of reliable information is found in museum specimens that were historically routinely collected in the past both for scientific purposes and as hunting trophies. These specimens constitute a reliable source of DNA for investigating the genetic structure of species or populations from the recent past [25,26,27]. Thus, the main aim of this study was to evaluate the status of the Italian grey partridge during the past century, particularly before the introduction of non-native individuals, using two different approaches: (i) collecting all the available literature on the releases of *P. perdix* in Italy since the beginning of the past century; (ii) characterizing the native genetic composition of Italian grey partridge through the analysis of ancient DNA extracted from museum samples.

## 2. Materials and Methods

### 2.1. Historical Documentation on Partridge Introduction and Release

The scientific library of the Institute for Environmental Protection and Research (ISPRA, Ozzano dell’Emilia, Italy) holds a historical collection of hunting magazines published by hunter associations since the beginning of the XX century. Bibliographic research was conducted on the ISPRA paper archive to obtain information on hunting activities and introductions of grey partridges by inspecting the most famous historical Italian hunting magazines and books in this field and searching for the keywords “starna” and “pernice” (common Italian partridges names).

From each document on restocking activities and trade of game birds, information such as capture locality, year, number of individuals released or hunted, and any other additional details were retrieved.

Using the keywords “starna” and “pernice”, overall, 308 documents containing text collection and images were identified and scanned, 143 of which focused specifically on grey partridge. Four categories were defined as follows:

Photo (PH): the pictures sent by the various sections of hunters are often not linked to the notes printed on the page; they also included a cartoon and a cover of Diana magazine.

Game launches (LC): game launches described in the regional hunters’ associations report.

Advertisement (PB): promotional materials of breeders, distributors, and importers.

Texts documents (TX): texts, articles, opinions, experiences about “game management”.

For each category, the number of documents retrieved is reported in Table 1.

Except for articles from the Italian Journal of Ornithology, information on species releases was found only in the grey literature, primarily in Italian hunting magazines. The examined time frames are based on the bibliographical materials housed at the ISPRA library of Ozzano dell’Emilia, Italy (Appendix A).

Most of these magazines had widespread distribution at the national level, except “Bologna venatoria” and “Liguria venatoria”, which were more focused at the local level, respectively, Central and Northern Italy, and reported communications collected mainly from local hunters’ associations (Appendix A).

### 2.2. Museum Sampling

Since early 2000, the principal museum curators and the National Association of Science Museums (ANMS) were contacted and asked about the possibility of collecting biological samples of *Perdix perdix* sampled in Italy (see Appendix A). The search for museum samples on a national scale was performed in a very wide way, also involving the national association of museums and contacting facilities throughout Italy. In addition, we created a network with the University of Perugia and the University of Piemonte Orientale in order to manage and share samples. We provided a detailed protocol for the collection of feathers, skin, and toe pads from museum specimens (see Appendix A) to allow the facilities interested in the project to perform a correct and standardized sampling that would allow viable DNA extraction. Thanks to these collaborations, 131 samples of grey partridge from the Italian peninsula were collected. Additionally, 9 museum samples from other countries, dated from 1948 to 1987 (Hungary *n* = 4, Czechoslovakia *n* = 1, former Yugoslavia *n* = 1, Poland *n* = 1, Scotland *n* = 2) were analyzed to gather information on the genetic patterns outside of Italy in the past century. The details on the origin of the complete dataset are reported in Table 2. Based on geographical origin, the samples were divided into three groups (Northern, Central, and Southern Italy). Samples from Emilia Romagna* were collected in the southern part of the region and thus considered in the Central Italy group (Table 2). Most museum specimens lacked exact geographical coordinates; therefore, we utilized the details available on the locality of the collection to arbitrarily choose approximate coordinates to show the territory coverage of the sampling (Figure 1). To highlight the relationships between European contemporary haplotypes, Italian museum and those of Liukkonen et al. [24], 141 sequences were downloaded from GenBank and used in statistical analyses (see Appendix A).

The article does not focus on currently extinct wild population but on the historical haplotypes of Italian grey partridge. This study started from museum specimens from the late 19th and early 20th centuries, a period when the relict Italian population was present, with reduced numerical consistencies, up to, in a few cases, the 1990s. Current individuals, both wild and bred, were deliberately excluded, as they were not informative for this study.

### 2.3. Genetic Characterization and Haplotype Identification

All the protocols were performed under a hood after the UV sterilization of plastics to reduce the risk of exogenous DNA contamination. DNA was extracted from feathers or skin tissues, using the commercial Qiagen DNeasy Blood and Tissue Kit, on the QIAcube robotic system (Qiagen), following the manufacturer’s instructions. A negative and a positive control were used to detect any DNA contamination event. All the different steps of the analyses (DNA extraction, pre-PCR, and post-PCR) were carried out in separate rooms to prevent the amplified PCR products from contaminating the earlier stages, and most of the liquid handling steps were performed by robots under a hood. For the same purpose, all the steps were carried out by working with a few samples at a time. The 550 bp long 5′ fragment of the mitochondrial control region (CR) was amplified utilizing the primer PHDL16750 (5′-AGGACTACGGCTTGAAAAGC-3′) and PH1H521 (5′-TTATGTGCTTGACCGAGGAACCAGA-3′) [28]. Amplicons were sequenced using the amplification primers and two additional internal primers [24]: PPE87F (5′-TCCCCATACATTATGGTAACAG-3′) and PPE316R (5′-GTACGTCGAGCATAACCAAA-3′). The PCR mix contained 10X reaction buffer, 0.2% BSA, 25 mM MgCl_2_, 2.5 mM DNTPs, 0.5 U Qiagen Hotstart Taq, and 10 mM of each primer, in a total volume of 10 μL. The thermocycler protocol PCR conditions started with an initial denaturation at 94 °C for 15”, followed by 40 cycles (94 °C × 40”, 55 °C × 40”, 72 °C × 40”), and ended with a final extension at 72 °C for 10 min.

Amplicons were purified with Exonuclease I and Shrimp Alkaline Phosphatase and sequenced with Big Dye v.3.1 terminator chemistry (Thermo Fisher Scientific, Waltham, MA, USA) following the manufacturer’s protocol. Sequencing was carried out on an ABI prism 3500 Genetic Analyzer (Thermo Fisher Scientific), and the resulting sequences were corrected with the software Seqscape v3.0 (Thermo Fisher Scientific).

When DNA amplification failed to produce amplicons due to low quantity or excessive fragmentation of the DNA, an internal shorter fragment of 170 bp was amplified using only the internal primers. Sequences were deemed reliable only when sequences from both forward and reverse primers had been obtained. Electropherograms were visually checked to control both the quality of the sequencing process and exclude the presence of double peaks [29].

An alignment of the sequences was produced using the Bioedit v7.0 software [30] and collapsed into single haplotypes in DnaSP v6 [31]. The haplotypes retrieved from the museum samples and analyzed in this work were compared with previously published data in GenBank^®^ (https://www.ncbi.nlm.nih.gov/ accessed on 5 September 2024) or obtained from Liukkonen and colleagues [24] (Appendix A).

### 2.4. Samples Subset Identification and Haplotype Description

The first evidence of the release of grey partridge in Italy dates to the beginning of the 20th century. From that period onward, the release events grew exponentially, particularly after the Second World War. According to this information, the sampling was divided into three temporal datasets: before 1915 (*n* = 54) considering the paper’s very low level of allochthonous introgression; between 1915 and 1945 (*n* = 29); and after 1946 (*n* = 57) to verify any changes in the genetic composition.

The number of haplotypes (N), haplotype (*Hd*) and nucleotide *(Pi*) diversity, and the number of polymorphic sites (S) were computed in DNAsp v6 [31] for the Italian museum samples.

Tajima’s D and Fu and Li’s were also computed in DNAsp to record any fluctuation in the studied populations. The haplotypes detected in this study, both belonging to Italian and European museum samples, were aligned in Bioedit v7.0 software [30] and compared with haplotypes downloaded from GenBank to verify any match with previous haplotypes described for the species [24,29,32,33,34,35]. A median-joining network [36] was constructed using PopART 1.7 software [37], describing the relationships (i) among Italian museum samples according to their geographical and temporal distribution and (ii) among contemporary and museum individuals in Italy and Europe.

## 3. Results

### 3.1. Data Retrieved from Historical Documentation on Partridge Introduction and Release

From the examination of the historical documentation, we obtained data on the operations of game introductions in Italy and focused on the areas of release, the date, the number, and the origin of individuals reintroduced (as detailed in Table 3).

In the documents, mainly in the advertisement sections, the presence of over fifty game farms and distributors operating in Italy was recorded. Most of them provided foreign specimens primarily from Central Europe (Hungary, Germany, Austria, Bulgaria, Romania, Denmark). The 136 text documents, dated from 1912 to 1985, dealt with the breeding, restocking, and management of huntable species, with a total of thirteen text documents focusing exclusively on grey partridges.

From all the documents collected, it was possible to record over three hundred episodes of grey partridge releases involving almost 90,000 individuals throughout the entire peninsula. A preponderance of introduction events in Northern Italy was observed, with more than 66,000 individuals in Piedmont; however, Central and Southern Italy also accounted for several reintroductions, with gross numbers of 20,000 and 2000 individuals, respectively.

Frequently, the origin of the reintroduced grey partridges (wild, captive bred, or the provenance localities) was missing, unless the indication of a foreign origin was occasionally reported. Several reintroduction events released individuals from Hungary (*n* = 15) and the Czech Republic (*n* = 13), with only occasional releases from Bulgaria, Poland, Austria, and Germany. On nine occasions, a few individuals were captured and released in Italy. These autochthonous releases were possible, as individuals were captured from game reserves called “bandite di caccia”, territories managed by state-owned entities, private owners, or hunters’ associations, where hunting was strictly forbidden and legally enforced to guarantee a safe environment for natural breeding of game species.

Massive restocking activities began after 1915, and before this date, only an introduction event was recorded in our bibliographic research in Italy. The first evidence of grey partridge introduction into the wild in Italy was recorded in 1913 and occurred in Lombardy, where 24 individuals were released from Bohemia. Subsequently, in the 1920s, four introductions were reported in Piedmont, Lombardy, and Veneto, consisting of around 250 individuals. Between 1928 and 1945, 110 introduction events documented indicate about 19,000 individuals released, covering the entire country. Massive introductions were made after the mid-20th century. In fact, after 1945 there were 86 introductions of 12,000 individuals in the 1950s, 72 introductions of 25,000 individuals in the 1960s, and 35 introductions of 30,000 individuals in the 1970s, for a total of 67,000 individuals in 193 releases (Table 3).

The analysis of the historical information (the grey literature) underlined a variation in both the frequence and consistency of the release of grey partridge in Italy. In particular, this practice has been documented since the beginning of the 20th century (first range: before 1915). In this period, animal releases were sporadic and numerically exiguous. Between 1915 and 1945, the release events grew exponentially (hundreds of releases), with an understandable interruption during the Second World War. After 1945, the release practice showed a marked and continuous increment (thousands of releases). For these reasons, and according to these data, the sampling was divided into three temporal datasets: before 1915 (*n* = 54), considering a very low level of allochthonous introgression; between 1915 and 1945 (*n* = 29); and after 1946 (*n* = 57) to verify any changes in the genetic composition. Furthermore, we chose to regroup the samples in clusters spaced by a few decades to avoid the creation of smaller and not statistically significant groups composed by few samples.

### 3.2. Museal Samples Genetic Characterization

Fifteen museum facilities and five private collections (see Appendix A) provided 131 samples collected in Italy between 1835 and 1999. Most of the samples originated from Northern (*n* = 38) and Central Italy (*n* = 69); due to the lack of museum samples and warmer climate conditions, which are less suitable for the species’ presence, only two samples were collected in Southern Italy. Moreover, 22 samples collected were excluded due to the absence of geographical referencing data.

Out of the 140 samples collected, 131 were from Italy and 9 from Europe, 37 failed to amplify, probably due to the DNA being too degraded from preservation procedures in taxidermy processes and conservation in museums, 20 presented evidence of contamination, and 9 showed unreliable genetic results, so they were removed from the following analysis. The remaining 74 samples produced 234 bp long sequences, ranging between nucleotides 96 and 329 of the mitochondrial control region, which includes the hypervariable region (Ref: GenBank^®^, *Perdix perdix* mitochondrion, complete genome NCBI reference sequence accession number: NC_039843.1). The Italian geographical distribution of reliable samples is shown in Figure 1. Unfortunately, the samples from Southern Italy did not produce reliable results, so it was not possible to assess the genetic composition of the Southern Italian population. Conversely, the analysis of all the nine European samples provided reliable sequences.

The sequences were collapsed into 15 different haplotypes, four of them being already deposited in GenBank (Table 4; Appendix A). The other 11 haplotypes were never published in any public biobank, and were deposited in GeenBank and provided of Accession Numbers (Appendix A). The majority of the 74 samples from Italy (*n* = 23) shared the haplotype Pdx_W1, while 17, 13, and 9 individuals were characterized, respectively, as haplotypes Pdx_W2, Pdx_W7, and Pdx_W5 (Table 4). Pdx_W1 was mainly distributed in Piedmont (n = 9; 39.1%) and throughout Northern Italy (*n =* 15; 65.2%), although a considerable percentage was also detected in Central Italy (*n =* 7; 30.4%). The haplotypes Pdx_W2 (*n =* 17 samples) and Pdx_W7 (*n =* 13 samples) were present mainly in Central Italy (*n =* 15; 88.2% and *n =* 12; 92.3%, respectively); only a few samples were collected in Northern Italy, respectively, *n* = 2 (11.8%) in Veneto and Piedmont, and *n* = 1 (7.7%) in Veneto. The haplotype Pdx_W5 (*n* = 9) was recorded only in Central Italy; the same applies for other eight low-represented haplotypes: Pdx_W6, Pdx_W8, Pdx_W9, Pdx_W10, Pdx_W12, Pdx_W13, Pdx_W14, Pdx_E1. Detailed information is shown in Table 4.

The samples collected before 1915 (*n* = 31) shared the haplotypes Pdx_W1 (*n* = 9), Pdx_W2 (*n* = 7), Pdx_W4 (*n* = 1), Pdx_W5 (*n* = 3), Pdx_W6 (*n* = 1), Pdx_W7 (*n* = 9), Pdx_W10 (*n* = 1).

The samples collected from 1915 to 1945 (*n* = 16), mostly shared the haplotypes Pdx_W2 (*n* = 5) and Pdx_W5 (*n* = 5), while the group of samples collected after 1945 shared mainly haplotypes Pdx_W1 (*n* = 13) and Pdx_W2 (*n* = 5) (Table 5). All nine museum samples from other countries presented haplotype Pdx_W1.

The haplotypes retrieved from the museum samples were mapped to GenBank^®^ using the BLAST function to verify species assignment and identify haplotypes already described and aligned with available sequences downloaded from GenBank^®^. All of them, excluding Pdx_W1, Pdx_E1, and Pdx_W3, were retrieved from the Italian samples (Appendix A and Table 4).

### 3.3. Genetic Variability

The 15 identified haplotypes showed a haplotype diversity (*Hd*) of 0.81 and a nucleotide diversity (*Pi)* of 0.011. A total of 22 polymorphic (S) and seven parsimony informative sites were described (Table 6; Appendix A).

The number of haplotypes and the haplotype diversity were similar across the three time-frames, although a slight reduction in genetic variability was documented in the individuals collected after 1945 (*Hd* = 0.738 and *Pi* = 0.009).

Tajima’s D computes low- or high-frequency polymorphisms relative to expectation; negative results document an excess of low-frequency polymorphisms, so they can be indicative of an expanding population, while positive values indicate an excess of both low and high-frequency polymorphisms and describe a decreasing population. Fu and Li’s tests are based on the distribution of haplotypes, and, if significant, they indicate demographic changes in the population or selection. After computing Tajima’s test, the null hypothesis of neutral evolution was confirmed: Tajima’s D was negative both in the subsample 1915–1945 and the whole sample dataset, even if not significantly so, and close to zero in the other two groups. Values from Fu and Li’s tests were not significant, except for the whole dataset, which was significantly negative (Table 6).

### 3.4. Genetic Structure and Phylogenetic Inference

The median-joining haplotype network (Figure 2) identifies the main widespread haplotypes and their connections, based on geographical distribution. Haplotypes Pdx_W5, 6, 7, 8, 9, 10, 12, 14 were recorded only south of the Po River, although Pdx_W7 were poorly represented (one sample) in a Northern region. Notwithstanding their position in the network and their connection with haplotypes mainly distributed in Southern Italy, Pdx_W4 Pdx_W11 and Pdx_W13 were recorded only in the north of the peninsula. The haplotype Pdx_W5 is directly connected to Pdx_W1 and is basal to Pdx_W7 and Pdx_W2. The haplotype Pdx_W2 is also connected to Pdx_W1 through Pdx_W12.

The same median-joining haplotype network (Figure 3) focused on temporal distribution. Seven haplotypes, Pdx_W1, Pdx_W2, Pdx_W4, Pdx_W5, Pdx_W6, Pdx_W7, and Pdx_W10, were present in group “ante 1915”, before the massive reintroductions; the presence of the haplotype Pdx_E1 in the group “from 1915 to 1945”, testifies the period of the effective start of releases with individuals from Eastern Europe.

Eight haplotypes were retrieved from the museum samples collected from 1946 to 1999, but only four of them (Pdx_W1, Pdx_W2, Pdx_W5, Pdx_W7) were shared with individuals from the two previous sampled periods.

A total of 141 sequences were downloaded from GenBank, collapsed into 47 haplotypes, and compared with the museum sequences (74 from Italy and 9 from Europe). A median-joining network was constructed to highlight the relationships between European contemporary and Italian museum haplotypes. Based on these integrated data, we determined the relationships between the haplotypes from GenBank, from Liukkonen et al. [24,32], and the museum samples from this study (Figure 4). Due to the short length of the fragment analyzed in the present study, several published haplotypes obtained from Liukkonen et al. [24] were indistinguishable from each other (W2-W28; W9-10-26; ME-E6-E14; E2-3-4). Haplotypes Pdx_W1 and Pdx_E1 matched the main haplotypes described by Liukkonen et al. 2002, respectively, MW and ME. Pdx_W2 was also recorded in the contemporary samples from 2000 in Northern Italy, the Pyrenees and an Italian farm (Haplotype W3, [24]). The Pdx_W3 was retrieved only in one sample outside Italy, in Denmark (Accession n. JN817442).

The Italian museum haplotypes were organized in two different phylogenetic lineages, both originating from the central haplotype Pdx_W1 (Figure 4). Haplotypes W2_28 and W21, Liukkonen et al. [24], positioned between Pdx_W1 and Pdx_W12, were retrieved, respectively, in Italy (W2 and W21) and in Finland (W28). Note that W2 and W28 haplotypes in the network correspond to a unique circle; however, considering the full length sequences in Liukkonen et al. [24], they are separated by four mutations.

## 4. Discussion

Wild game bird species, and particularly Galliformes, are threatened by management policies that are aimed at sustaining populations to support hunting demands [15,16]. Among them, the grey partridge has been an important European game species for centuries and has therefore been widely involved in and impacted by anthropogenic translocations and stocking efforts.

The analysis of the mitochondrial control region of Italian and European museum samples, as well as the survey of the available grey literature on partridge management in Italy, allowed us to gain new insights into the genetic diversity and composition of Italian grey partridge, before and after the massive releases of individuals carried out since the middle of the past century. The stocking of native populations with farm-bred individuals from allochthonous populations may have had adverse genetic effects on local wild populations through introgressive hybridization, leading to loss of genetic diversity [38]. The article does not focus on the currently extinct wild population, but rather on the historical haplotypes of Italian grey partridge, starting from museum specimens from the late 19th and early 20th centuries, a period when the relict Italian population was present, with reduced numerical consistencies, up to, in a few cases, the 1990s. We deliberately excluded current individuals, both wild and bred, as they were not useful in this study.

Understanding the original genotypic asset then represents a fundamental starting point, not just to define the population dynamics over the last century, but also to map the expansion or contraction of a genetic pool, as well as the presence of alien genotypes. The evidence gathered in this work highlights the presence of six unique haplotypes in Italian museum samples dated before 1915, which were never detected in any other populations. Three of these haplotypes, Pdx_W4, Pdx_W6, and Pdx_W10, were detected only in the samples from individuals collected before 1915. These haplotypes were never detected again in the other two time-frame groups of samples, suggesting that these haplotypes were removed from the gene pool after the start of the restocking efforts. The remaining four haplotypes, Pdx_W1, Pdx_W2, Pdx_W5, and Pdx_W7, were collected in all three time-frame groups, despite the massive introduction events. However, due to the limited availability of museum samples, it cannot be entirely excluded that these results could be biased and that the same haplotypes were present in the three timeframes. Apart from this consideration, among them, Pdx_W1 was shared and is still common today in the Western European population. The haplotypes more represented after Pdx_W1 were, in order of abundance, Pdx_W2, Pdx_W7, and Pdx_W5, detected principally in Tuscany and Emilia Romagna. Before 1915, the haplotype Pdx_W2 was detected only in Tuscany and Emilia Romagna; conversely, after 1915 it was detected from Piedmont and Veneto through Emilia Romagna and Tuscany, reaching Latium and Marche, being widespread across Northern and Central Italy. Despite these peculiarities, no strong differences were found in the composition of the metapopulation, subdividing it into three time-frame groups (1835–1914, 1915–1945, and 1946–1999), as highlighted in Figure 3.

On the contrary, the metapopulation seems subdivided at the geographical scale, as already reported, following the distinction among the PW and PE haplotypes [24]. The Italian genetic pool, with high detected variability, the presence of unique haplotypes and a subspecies, allowed us to reasonably hypothesize that the peninsula represented a glacial refugium for the grey partridge, and that the Alps acted as a natural barrier, able to isolate the Italian population, promoting the fixation of adaptive traits that led to the formation of the Italian grey partridge subspecies [39]. The structure of the Italian population, obtained from museum haplotypes, does not present a clear division into geographical areas (Figure 2), such as north versus central Italy, although there are haplotypes found only in the north (Pdx_W3, 4, 11, 13), only in the center (Pdx_W5, 6, 8, 9, 10, 12, 14), or predominantly in one of the two regions (Pdx_W1 predominantly central, Pdx_W4 and 7 predominantly northern). 

Italian museum samples showed the presence of the main Western European haplotype Pdx_W1 (formerly MW; see Appendix A) [24], which is basal for the clade and shared with many other EU populations, including France, Spain, Bulgaria, Poland, Germany, Sweden, England, Ireland, Latvia, Finland [24], from which two different phylogenetic Italian lineages originated (Figure 4). The haplotypes Pdx_W6 and Pdx_W3, which originate directly from Pdx_W1 and Pdx_W8, which derives from W14 (Germany), remain excluded from these two phylogenetic lineages. It can also be noted that the haplotypes from central Italy (Pdx_4, Pdx_11, Pdx_13) within the two Italian phylogenetic lineages, are always basal with respect to the haplotypes retrieved only from the Italian Northern areas, thus supporting the existence of isolation processes during the Pleistocene ice age.

Eight (Pdx_W3, Pdx_W8, Pdx_W9, Pdx_W11, Pdx_W12, Pdx_W13, Pdx_W14, and Pdx_E1) of the 15 haplotypes detected in the Italian museum samples were reported only after 1915, even in areas that were also sampled before 1915. All these haplotypes were sampled once, whereas Pdx_W12 was sampled twice. This could be explained by a lack of sampling before 1915, or it could be due to reintroductions with allochthonous haplotypes no longer found in contemporary individuals. Not surprisingly, most Italian haplotypes are not found in other European populations, supporting the hypothesis that the Italian peninsula served as a glacial refugium where the *P. p. italica* was differentiated. As already documented for several species, including Fasianidae [40], the Pleistocene Ice Age deeply influenced species distribution and diversification. The isolation of single species in allopatric refugia during long glacial periods was alternated by repeated expansion phases, establishing hybrid zones upon secondary contact and being the major cause of intraspecific divergence in temperate species [41]. Furthermore, the role of phylogeographical lineage divergence due to Pleistocene refugia in producing post-glacial variation is documented at both ecotypic and subspecific levels, as seen in *P. p. italica* [23].

The haplotype Pdx_E1 is a basal and widespread haplotype in Eastern European populations [24]. The presence of the main common haplotype of Eastern European populations in the Italian samples dated after 1915, confirms that reintroductions with individuals from these territories were probably a well-established practice in those years, as also documented by the data extracted from the grey literature of the time, examined in this work.

The molecular approach to species, subspecies, and significant evolutionary units (ESUs) can provide insights into their phylogeography and historical genetic population composition and offer valuable support to management species conservation actions [40].

The high values of haplotype diversity and nucleotide diversity that were found in each time-range group, when compared to other similar studies (e.g., Andersen et al., 2011 [34]), could be attributed to the repeated and continuous restocking actions using individuals from different regions and other subspecies. In particular, more than one century of introductions of grey partridges in Italy, along with the severe decline of the Italian subspecies, has left only relict traces of the historical indigenous gene pool in the Italian peninsula [16,41], underlining genetic, ecological, and ethological differences between wild and introduced individuals. After all, the genetic differentiation between individuals used in restocking and the wild populations was already reported for other *Perdix* subspecies, such as *P. p. hispaniensis* in the Pyrenees [10] or *P. p. perdix* in Denmark [34].

Indeed, the genetic data obtained since the early 2000s were fundamental for Project Life *Perdix*—Life17 NAT/IT/000588, particularly for the breeding selection aimed at the reintroduction in nature of the Italian grey partridge in the Mezzano Valleys (Ferrara province). Individuals characterized by haplotypes, historically present in Italy were utilized as the breeding pool for the restocking actions of the project. This information is mandatory in restocking plans (both for hunting purposes and especially for conservation purposes), mainly for species with low dispersal abilities and marked differentiation throughout the distribution area to avoid outbreeding depression and introgression among different subspecies [42].

The greatest challenge we faced in implementing the LIFE project was establishing a reference timeline and determining which haplotypes were autochthonous before the massive introduction of non-native individuals. After that, it was necessary to find living individuals with these characteristics for breeding, maintaining the necessary variability despite starting with a limited number of individuals. The fact that the LIFE project is now in its final year and that thousands of genetically selected individuals, in terms of both variability and affiliation with native lines, have been released, is a proof of how these difficulties were overcome. It is still too early to determine whether these efforts have yielded measurable conservation results for the species in Italy. However, the initial data are encouraging. The monitoring of the specimens shows the persistence of released family groups in areas where they had previously disappeared. In the near future, we will be able to have numerical data that will allow for a complete analysis of the project’s success.

While one of the main limitations of this study is the length of the analyzed fragments and the impossibility of obtaining DNA of sufficient quality from some samples, it is to be considered that the sampled analyzed underwent not just the hardship of time, but also several harsh processes required to taxidermize specimens. Furthermore, most taxidermists in the past centuries used arsenic compounds [43] or other sample preservation methods [44,45] during the tanning process to stop the decaying process. Unfortunately, these compounds can also act as PCR inhibitors, hindering the possibility to carry out molecular studies on ancient taxidermized specimens [45]. Another possible limitation could be opportunistic sampling, based on the only available well-preserved specimens in museums, which could lead to an underrepresentation of certain haplotypes. However, but it is important to take in account the nature of the analyzed samples. Historic specimens, dated before the beginning of the 20th century, are not just hard to find, but they are also particularly difficult to process for molecular analyses.

The integration of data from museums, biobanks of historical biodiversity, and bibliographic sources allowed the creation of a transversal dataset useful for evaluating genetic variability loss. In particular, the bibliographic collection of grey literature was fundamental to gaining information never retrieved before regarding the extent of the releases, and their relative spatial and temporal distribution, accounting for hundreds of releases of tens of thousands of individuals all over the Italian peninsula. The number of releases and, even more, the number of individuals involved grew exponentially from the beginning of 1900 to the end of the century. According to the result of the bibliography research, the first release event recorded was carried out in 1913, but it is not possible to exclude the occurrence of unreported introduction events, especially if carried out on smaller scales and at regional levels. The integrative analysis of museum specimens and grey literature represents the only way to reconstruct the history of such a profoundly manipulated species. The historical data obtained from the review of grey literature helps us determine up to what period we can consider the haplotypes as representative of the Italian population. It also confirms that the dates identified as the beginning of the massive introduction of non-native individuals coincide with the discovery of haplotype Pdx_E1 in an animal, which is consistent with those dates.

## 5. Conclusions

This research was conducted by combining two different methods. The literature review of historical collections and hunting magazines published by hunter associations since the beginning of the 20th century, housed in the scientific library of the Institute for Environmental Protection and Research (ISPRA, Ozzano dell’Emilia, Italy), allowed the identification of the dates, locations, and quantities of grey partridge releases from the late 1800s to the late 1900s. Three historical periods, characterized by releases in the order of tens (pre-1915), hundreds (1915 to 1945), and thousands (post-1945) were identified.

A molecular analysis was then conducted on museum specimens from Italy, dating back to the late 1800s, using the mitochondrial control region to identify grey partridge haplotypes that could help understand the dynamics of the gene pool in the region. Fifteen different haplotypes were identified, 11 of which had never been published before. Some of these are present exclusively before 1915, while others appear in later periods, such as Pdx_E1, typical of Eastern European populations, indicating the partial replacement of the original gene pool due to the introduction of non-native individuals. It appears difficult from the network analysis to obtain a correlation between haplotypes and historical periods or the geography of the Italian peninsula (e.g., north versus south). The Italian museum haplotypes were mostly organized into two different phylogenetic lineages, both originating from the central haplotype Pdx_W1. The presence of unique haplotypes and a subspecies allows us to reasonably hypothesize that the peninsula represented a glacial refugium for the grey partridge, promoting the fixation of adaptive traits.

One of the main limitations of this study is the length of the analyzed fragments due to processes required to taxidermize specimens, which can also act as PCR inhibitors. Another possible limitation could be the sampling, which was based solely on the available well-preserved specimens in museums, which could lead to an underrepresentation of certain haplotypes. The present research underlined the uncertain future of grey partridge in Italy and the importance of the major conservation efforts already performed. With an extremely rarefied distribution and a high number of individual releases occurring in the past decades and still ongoing, for hunting purposes, the future of the Italian grey partridge is still in jeopardy. For these reasons, the obtained data will be of crucial importance for future research and management plans. The molecular characterization of historical Italian populations from up to 1999 and the comparison and network with European data represent a key step toward conservation efforts. In fact, for the first time, these data will be a fundamental reference for future studies, whether based on historical or actual specimens. Furthermore, these data will also drive direct breeding plans to carry out restocking interventions with native genotypes in the framework of management projects. The Life Perdix—Life17 NAT/IT/000588 is a project representing an example of how information derived from historical data could be used and applied for conservation and management purposes. The preservation of DNA from museum specimens does not prevent the disappearance of specific genotypes, but it can compensate for the great loss of knowledge that accompanies the extinction of animal species. In particular, the genotyping of specimens from museum collections does not only represent cognitive data in itself but also provides indications on the modern genotypes most similar to the historical ones to be used for any restocking actions. In particular, in the case of the grey partridge present in Italy, we highlighted that it exhibited unique haplotypes not found in European populations. Therefore, in the event of reintroductions, it will be important to consider these factors, in addition to ensuring the necessary genetic variability. The implementation of the LIFE project posed a series of difficulties and challenges (establishing a timeline, identifying native haplotypes, maintaining variability in breeding) that were addressed and resolved, culminating in the release of thousands of individuals into a territory prepared with environmental improvements to guarantee survival. The initial monitoring data show the persistence of breeding pairs and groups of individuals in areas where they had previously disappeared. Long-term data will be necessary to determine if the conservation effort will yield the desired results.

Another crucial objective of this work, in addition to identifying and publishing the haplotypes exclusive to the Italian territory before the massive introductions, was to clarify and unify all the haplotypes publicly available in GenBank by assigning them a unique identifier, with the aim of laying the basis for future research studies. Evidence-based restoking efforts will promote the preservation and restoration of native and autochthonous genetic traits that evolved over centuries in the Italian peninsula and that were almost entirely wiped out by the stocking of allochthonous individuals, not just the Italian grey partridge, but also several other threatened Italian endemic species.

## Figures and Tables

**Figure 1 biology-13-00709-f001:**
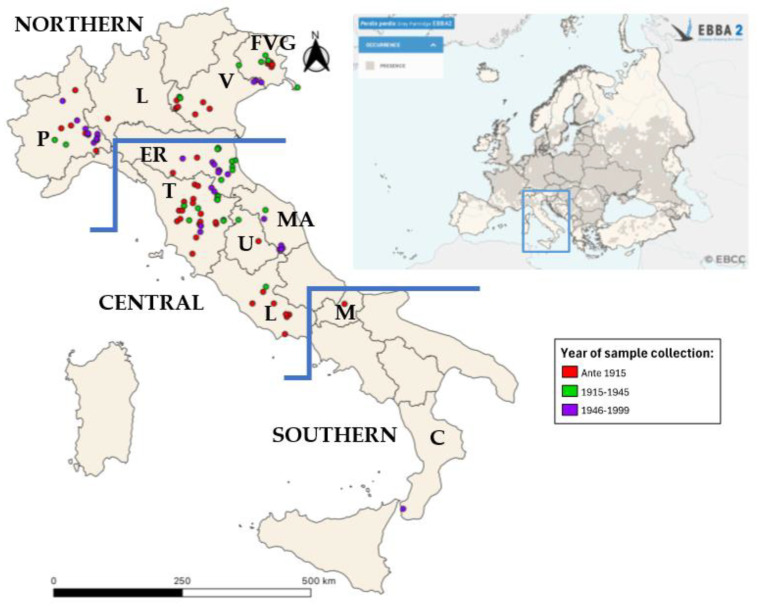
Geographic distribution of sampling and of Perdix perdix in Europe. Geographical distribution of the analyzed samples in the Italian peninsula (left). Samples were color-coded based on the collection year: samples collected before 1915 are represented in red; samples collected from 1915 to 1945 are represented in green; samples collected from 1946 to 199 are represented in purple. A map from EBBA2 of the occurrence of Perdix perdix in Europe is presented in the upper right corner [14].

**Figure 2 biology-13-00709-f002:**
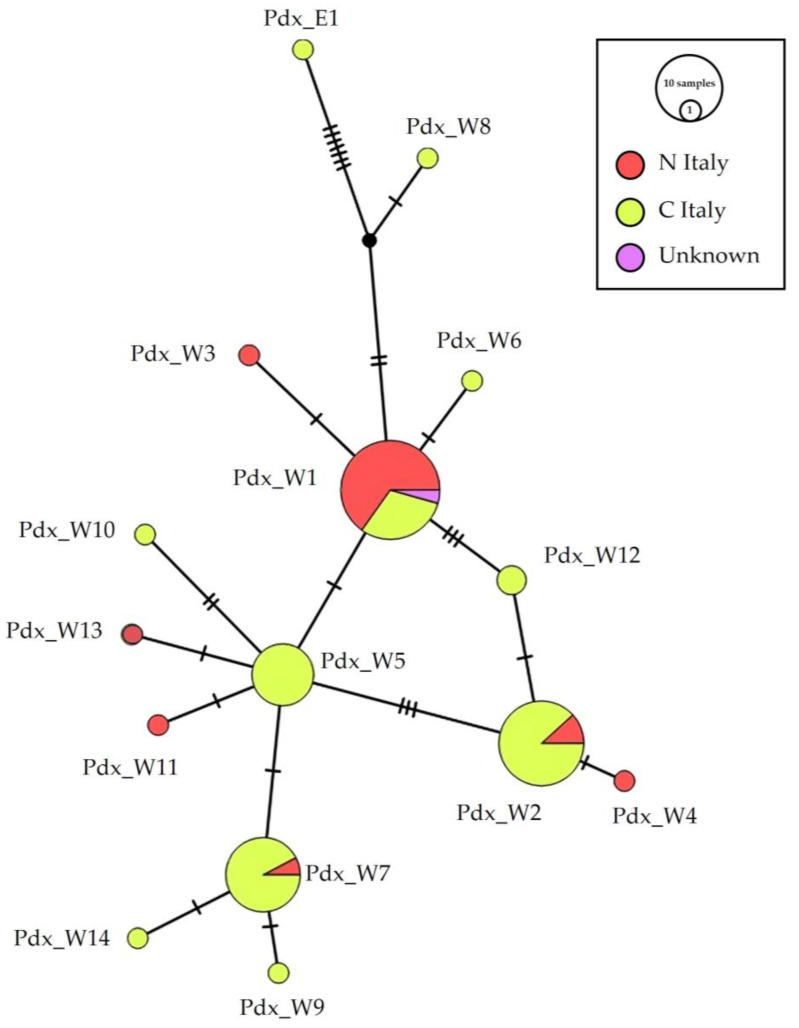
Network of the geographical distribution of haplotypes. Median-joining network of the Italian P. perdix CR haplotype detected, showing the spatial distribution of the analyzed samples: in orange are samples from individuals collected in Northern (N_Italy) and in yellow are samples from individuals collected in Central Italy (C_Italy), in pink are samples from individuals with unknown origin of collection. The size of the circles is proportional to the number of samples. Each line between two crossbars or two circles indicates one mutation. Pdx_W = western, Pdx_E = eastern. For details on mutation and position see Appendix A.

**Figure 3 biology-13-00709-f003:**
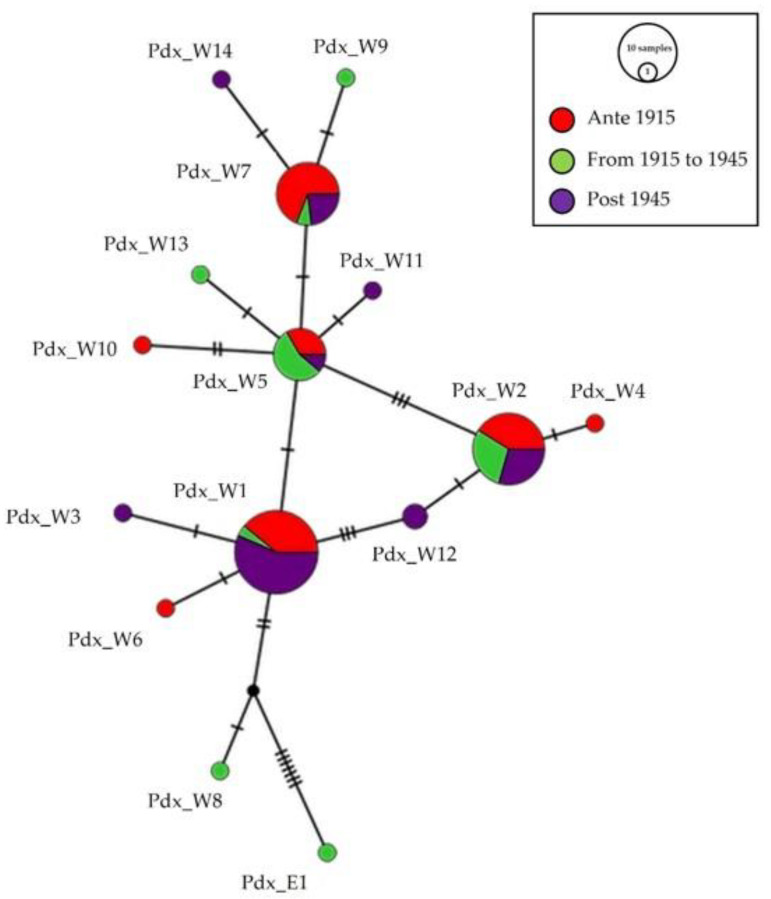
Network of the temporal distribution of haplotypes. Median-joining network of Italian P. perdix CR haplotypes, showing the temporal distribution of the analyzed samples: in red are samples from individuals collected before 1915, in green are samples from individuals collected between 1915 and 1945, and in purple are samples from individuals collected after 1945. The size of the circles is proportional to the number of samples. Each line between two crossbars or two circles indicates one mutation. Pdx_W = western, Pdx_E = eastern.

**Figure 4 biology-13-00709-f004:**
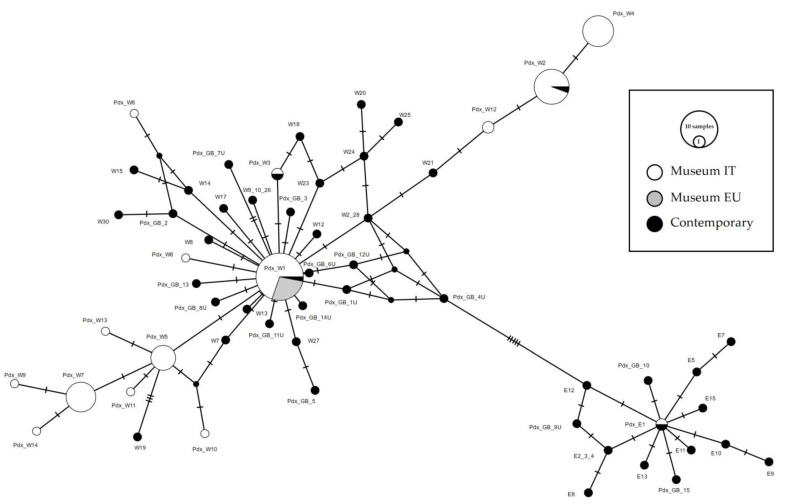
Median-joining network of Perdix perdix CR haplotypes from European samples. The Italian museal samples obtained from this study are represented in white, the European museal samples obtained from this study are represented in grey, whereas the contemporary samples retrieved from GenBank are represented in black. The size of the circles is proportional to the number of samples. Each line between two crossbars or two circles indicates one mutation. Pdx_W = western, Pdx_E = eastern.

**Table 1 biology-13-00709-t001:** Subdivision in the four categories of grey literature collected. The table shows the different types of documents considered and their relative numerical consistency.

	Photo (PH)	Game Launches (LC)	Advertisements (PB)	Text Documents (TX)
Document number	24	105	43	136

**Table 2 biology-13-00709-t002:** Geographical origin of the samples. They were collected from 15 museum facilities and 5 private collections. Samples from Emilia Romagna* were collected in the Southern part of the region and thus considered in the Central Italy group.

Country	Geographic Location	N.	Italian Region	N
Italy	Northern	38	Piedmont (P)	17
			Veneto (V)	10
			Lombardy (L)	1
			Friuli Venezia Giulia (FVG)	10
	Central	69	Emilia Romagna* (ER)	19
			Marches (MA)	4
			Tuscany (T)	32
			Latium (L)	8
			Umbria (U)	6
	Southern	2	Molise (M)	1
			Calabria (C)	1
	Italy	22	Unknown region	22
Total Italian samples				131
Europe		9	Hungary	4
			Scotland	2
			Czechoslovakia	1
			Former Yugoslavia	1
			Poland	1
Total samples from other countries				9
Total samples				140

**Table 3 biology-13-00709-t003:** Number and time of releases and individuals all over the country from 1913 up to 1971. One row for each region, subdivided into North, Central, and South Italian peninsula, reporting the following: the number of release events, the number of individuals released, and the time frame, considering the first and last year of recorded release.

Geographic Location	Region	N° Release	N° Individuals	Time Frame
North Italy	Friuli-Venezia Giulia	9	43	1930–1965
	Liguria	52	10,217	1930–1971
	Lombardy	60	9024	1913–1971
	Piedmont	46	34,378	1932–1971
	Trentino-Alto Adige	3	12	1931–1959
	Veneto	37	3476	1928–1970
	Total	411	57,150	
Central Italy	Abruzzo	5	906	1966–1971
	Campania	14	2436	1932–1970
	Emilia-Romagna	43	9135	1932–1969
	Lazio	13	4476	1930–1960
	Marches	8	3161	1932–1971
	Tuscany	44	6849	1951–1970
	Umbria	17	1973	1939–1969
	Total	144	28,936	
South Italy	Basilicata	3	106	1955–1970
	Calabria	5	628	1950–1971
	Molise	2	410	1942–1965
	Apulia	5	310	1932–1963
	Sicilia	3	415	1932–1970
	Total	18	1869	1913–1971

**Table 4 biology-13-00709-t004:** Distribution of haplotypes in Italy. Only one sample was of unknown Italian origin and had haplotype Pdx_W1. Haplotypes have been deposited in GenBank (Appendix A). * Accession numbers are reported in the last column.

Haplotype	Italy	Northern Italy	Central Italy	Other Countries	Accession Number
Pdx_W1 *	23	15	7	9	PQ299011
Pdx_W2 *	17	2	15		PQ299012
Pdx_W3 *	1	1			PQ299013
Pdx_W4	1	1			PQ299014
Pdx_W5	9		9		PQ299015
Pdx_W6	1		1		PQ299016
Pdx_W7	13	1	12		PQ299017
Pdx_W8	1		1		PQ299018
Pdx_W9	1		1		PQ299019
Pdx_W10	1		1		PQ299020
Pdx_W11	1	1			PQ299021
Pdx_W12	2		2		PQ299022
Pdx_W13	1	1			PQ299023
Pdx_W14	1		1		PQ299024
Pdx_E1 *	1		1		PQ299025
	74	22	51	9	

**Table 5 biology-13-00709-t005:** Distribution of haplotypes in Italy in the three time-frames. The choice of threshold was due to information on the first releases (1915), and the starting of massive reintroductions (after the Second World War). Each color indicates a haplotype identified only in a specific time frame (red: before 1915, green: between 1915–1945, blue: after 1945).

Haplotype	Italy	1835–1914	1915–1945	1946–1999
Pdx_W1	23	9	1	13
Pdx_W2	17	7	5	5
Pdx_W3	1			1
Pdx_W4	1	1		
Pdx_W5	9	3	5	1
Pdx_W6	1	1		
Pdx_W7	13	9	1	3
Pdx_W8	1		1	
Pdx_W9	1		1	
Pdx_W10	1	1		
Pdx_W11	1			1
Pdx_W12	2			2
Pdx_W13	1		1	
Pdx_W14	1			1
Pdx_E1	1		1	
Haplotypes n°	15	7	8	8
Samples n°	74	31	16	27

**Table 6 biology-13-00709-t006:** Variability indices in the three temporal groups. The asterisks indicate the statistical significance of the observed value of the indices, where no asterisk means not significant, one asterisk low significant up to three asterisks highly significant. Two asterisks identified significant value, with *p*-value < 0.02.

Variability Indices	1835–1914	1915–1945	1945–1999	Total
Haplotypes Number (N)	7	8	8	15
Polymorphic sites (S)	9	15	8	22
Haplotype diversity (*Hd*)	0.807± 0.037	0.833 ± 0.070	0.738 ± 0.076	0.813 ± 0.025
Nucleotide diversity (*Pi*)	0.010	0.015	0.009	0.011
Tajima’s D	0.012	−1.060	0.16	−1.4
Fu and Li’s D test	−0.914	−1.141	−0.520	−4.184 **
Fu and Li’s F test	−0.701	−1.289	−0.369	−3.771 **

## Data Availability

All sequences were submitted online to GenBank, and the Accession numbers is provided in Table 4 and Appendix A.

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
