# Peer review of "Combining Historical and Molecular Data to Study Nearly Extinct Native Italian Grey Partridge (Perdix perdix) at the Turn of the Twentieth Century"

_biology, 2024, doi:10.3390/biology13090709_

Round 1

Reviewer 1 Report

Comments and Suggestions for Authors

The main question addressed by the research group is to study the status of the Italian grey partridge using two ways: 1. Exploring the past literature and 2. Characterization of the native genetic composition through the analysis of DNA samples extracted from museum. The conclusion is well written, and it is covering the questions that needs to be addressed. The conclusion could be more descriptive. Also, what challenges or obstacles could be faced in implementing the conservation strategies could be described in the conclusion. The references are appropriate. The manuscript is well-written, and the text is clear and easy to read. The molecular data mentioned in this manuscript makes it original as compared to the other literature on this extinct bird. The title could be modified by introducing ‘extinct’ to it. All the figures and fonts need formatting. Some images of the bird would be nice to have while showing the haplotype differences. How is the success of conservation efforts monitored and measured could be added. Overall, really good manuscript.

Author Response

RESPONSES TO REVIEWER 1

Thank you for your suggestions which we appreciated. Below we have inserted the responses to your observations and have tried to modify the text of the work according to your indications.

Comment 1

The conclusion could be more descriptive.  Also, what challenges or obstacles could be faced in implementing the conservation strategies could be described in the conclusion.

Response to comment 1

we add to discussion section the following text at line 570:

“The greatest challenge we faced in implementing the LIFE project was establishing a reference timeline and determining which haplotypes were autochthonous before the massive introduction of non-native individuals. After that, it was necessary to find living individuals with these characteristics for breeders, maintaining the necessary variability despite starting with a limited number of individuals. The fact that the LIFE project is now in its final year and that thousands of genetically selected individuals, in terms of both variability and belonging to native lines, have been released, is a proof to how these difficulties were overcome. It is still too early to determine whether these efforts have yielded measurable conservation results for the species in Italy. However, the initial data are encouraging. The monitoring of the specimens shows the persistence of released family groups in areas where they had previously disappeared. In the near future, we will be able to have numerical data that will allow for a complete analysis of the project's success.”

And to conclusion section the following text at line 613:

“This research was conducted by combining two different methods. The literature review of historical collections and hunting magazines published by hunter associations since the beginning of the 20th century, housed in the scientific library of the Institute for Environmental Protection and Research (ISPRA, Ozzano dell’Emilia, Italy), allowed the identification of the dates, locations, and quantities of grey partridge releases from the late 1800s to the late 1900s. Three historical periods characterized by releases in the order of tens (pre-1915), hundreds (1915 to 1945), and thousands (post-1945) were identified.

A molecular analysis was then conducted on museum specimens from Italy dating back to the late 1800s, using the mitochondrial control region to identify grey partridge haplotypes that could help understand the dynamics of the gene pool in the region. Fifteen different haplotypes were identified, 11 of which had never been published before. Some of these are present exclusively before 1915, while others appear in later periods, such as PdxE1, typical of Eastern European populations, indicating the partial replacement of the original gene pool due to the introduction of non-native individuals. It appears difficult from the network analysis to obtain a correlation between haplotypes and historical periods or the geography of the Italian peninsula (e.g., north versus south). Italian museum haplotypes were mostly organized into two different phylogenetic lineages, both originating from the central haplotype Pdx_W1. The presence of unique haplotypes and a subspecies reasonably allows us to hypothesize that the peninsula represented a glacial refugium for the grey partridge, promoting the fixation of adaptive traits.

One of the main limitations of this study is the length of the analyzed fragments due to processes required to taxidermize specimens, which can also act as PCR inhibitors. Another possible limitation could be the sampling, based solely on the available well-preserved specimens in museums, which could lead to an underrepresentation of certain haplotypes.…..

….. The implementation of the LIFE project posed a series of difficulties and challenges (establishing a timeline, identifying native haplotypes, maintaining variability in breeding) that were addressed and resolved, culminating in the release of thousands of individuals into a territory prepared with environmental improvements to guarantee survival. The initial monitoring data show the persistence of breeding pairs and groups of individuals in areas where they had previously disappeared. Long-term data will be necessary to determine if the conservation effort will yield the desired results.

Another crucial objective of this work, in addition to identifying and publishing the haplotypes exclusive of the Italian territory before the massive introductions, was to clarify and unify all the haplotypes pubblically available in GenBank by assigning them a unique identifier, with the aim of laying the basis for future research studies.”

Comment 2

The references are appropriate.  The manuscript is well-written, and the text is clear and easy to read. The molecular data mentioned in this manuscript makes it original as compared to the other literature on this extinct bird.

The title could be modified by introducing ‘extinct’ to it.

Response to comment 2

We changed the title in this direction. Considering that there is a Life project underway and some relict populations, it seemed too strong to write extinct. Now the title is “Combining historical and molecular data to study nearly extinct native Italian grey partridge (Perdix perdix) at the edge of the twentieth century.”

Comment 3

All the figures and fonts need formatting.

Response to comment 3

We have formatted the tables and figures where necessary (changes in the text of the work in red).

Comment 3

Some images of the bird would be nice to have while showing the haplotype differences.

Response to comment 3

We seriously have considered this suggestion. However, the morphological differences are still subject of scientific debate today and are small, such as "less rusty or reddish color in the upper parts, rump with dark brown bars, rather than reddish".  We believe that these small differences would difficulty be noticeable in small overall photo, with the risk of misinterpretation. For these reasons we prefer to not include animals’ photos.

Comment 4

How is the success of conservation efforts monitored and measured could be added.

Response to comment 4

We addressed this issue in first answer by modifying the discussion and conclusions.

Reviewer 2 Report

Comments and Suggestions for Authors

This is an interesting study that looks at how the genetic makeup of quail, a game bird, has changed over the past 100 years. It will be an example of many birds around the world that have been artificially bred and released.

L88

"limited insect availability due to pesticides," It would be better to write more clearly. What role do insects play here?

Table1.

I want Italian place names and map notations to show them. People from other countries would not know which is the south and which is the north just by looking at the place names.

L180

Misspelling.

Result 3.1

Most of the information here is in the method section, so it's better to move it.

Table 3 is not the main result, so I think it should be treated as an appendix.

Table 4. Use commas instead of periods

L270

"Central Europe" - which country specifically is this? The country of origin is very important, as it is from which country the individuals were bred and released.

discussion

The fact that the E haplotype, which is the main one in Eastern Europe, was also detected in Italy is evidence of genetic contamination, so I think it would be better to consider this in more depth.

As a result, these haplotypes have disappeared and new haplotypes have appeared, so what is the current genetic structure of quails? This is the most interesting point of this paper, so I think the authors should address this question. 

Author Response

Responses to Reviewer 2

Thank you for your suggestions which we appreciated. Below we have inserted the responses to your observations and have tried to modify the text of the work according to your indications.

Comment 1

L88

"limited insect availability due to pesticides," It would be better to write more clearly. What role do insects play here?

Response 1

The animal component (small invertebrates) is of considerable importance in the first three weeks of life of chicks and during the breeding phase. In spring and summer, the diet is enriched with ground insects and Annelids, especially appetite are the larvae of Earthworms, Lepidoptera and Orthopters.

Therefore, we have changed the draft to the line 88 adding “fundamental in the diet of chicks for the first months of life and during the breeding phase”

Comment 2

Table1.

I want Italian place names and map notations to show them. People from other countries would not know which is the south and which is the north just by looking at the place names. 

Response 2

We have modified Figure 1 and Table 1 according to your requests

Comment 3

L180

Misspelling.

Response 3

On line 213 we changed the word Quiagen to Qiagen.

Comment 4

Result 3.1

Most of the information here is in the method section, so it's better to move it. 

Response 4

We have seriously evaluated this comment for which we thank the reviewer. After a deep analysis of the text, following your suggestion, we have modified the manuscript moving the part from line 229 (first version) to 145 (new version) of the first draft version into the materials and methods section. The rest of the results section is related to the results obtained from reading the material, in particular the number of releases, dates of releases, number of individuals and places of releases.

Comment 5

Table 3 is not the main result, so I think it should be treated as an appendix.

Response 5

We have moved Table 3 to the appendix section

Comment 6

Table 4. Use commas instead of periods 

Response 6

We have replaced periods with commas in Table 4

Comment 7

L270

"Central Europe" - which country specifically is this? The country of origin is very important, as it is from which country the individuals were bred and released. 

Response 7

We added at lines 284 the countries mentioned in the advertisements in brackets after "Central Europe": “In the documents, principally in the advertisement sections, the presence of over fifty game farms and distributors operating in Italy was recorded, most of them provided foreign specimens principally from Central Europe (Hungary, Germany, Austria, Bulgaria, Romania, Denmark)”.

Comment 8

Discussion

Comment 8.1

The fact that the E haplotype, which is the main one in Eastern Europe, was also detected in Italy is evidence of genetic contamination, so I think it would be better to consider this in more depth.

Response 8.1

We did not emphasize this point in the discussion, so we added a comment about it in the line 544 as reported below.

“The presence in Italian territory of the haplotype main common of the populations of Eastern Europe, in the samples dated after 1915, confirm that reintroductions with individuals from these territories were probably a well-established practice in those years, as also documented by the data extracted from the grey literature of the time examined in this work.”

Comment 8.2

As a result, these haplotypes have disappeared and new haplotypes have appeared, so what is the current genetic structure of quails? This is the most interesting point of this paper, so I think the authors should address this question.

Response 8.2

The article focuses not on current data but on the characterization of the Italian historical haplotypes, starting from museum specimens from the late 19th and early 20th centuries, a period when the Italian population was small but still present in the territory, up to, in a few cases, the 1990s. The aim was to investigate, if possible, the genetic makeup of the Italian grey partridge before the massive reintroductions that began in the 1920s. The conservation goal is to provide scientific bases for managing and conservation actions for the taxon.

According to official reports (Article 12 of the Birds Directive), the taxon is currently considered extinct in the wild in Italy (Nardelli et al., 2015), and the continuous introductions for hunting purposes with non-native individuals only further dilutes any remaining traces of the past original genetic structure.

According to a 2014 estimate, the relict surviving populations (which are at high risk of quickly disappearing due to stochastic events) were limited to only a few areas in Val cerrina (Alessandria) northern and Gran Sasso central Italy, while in southern Italy, the species is extinct. (Trocchi V.; Riga F.; Meriggi A.; Toso S. Piano d’azione Nazionale per La Starna (Perdix perdix); Conservazione della Natura, Ed.; Quaderni, 39.; ISPRA, 2016;). Outlining the genetic structure of the grey partridge in Italy it is not possible as the population no longer exists, having been replaced by almost one hundred years of introductions with allochthonous individuals.

We added a comment in the lines 210 as reported below.

“The article does not focus on current extinct in the wild population, but on the historical haplotypes of Italian grey partridge. The study started from museum specimens from the late 19th and early 20th centuries, a period when the relict Italian population was present with reduced numerical consistencies, up to, in a few cases, the 1990s. Current individuals, wild and breed, were deliberately excluded as they were not informative for this study.”

Reviewer 3 Report

Comments and Suggestions for Authors

The paper focuses on the genetic characterization of the grey partridge (Perdix perdix), specifically targeting the Italian subspecies. The study utilized the mitochondrial control region (CR) to identify haplotypes, comparing pre- and post-1915 populations to understand the impact of anthropogenic pressures on the genetic diversity of the species. The findings revealed a significant decline in genetic diversity, with several haplotypes disappearing after 1915 and new haplotypes emerging in later populations, likely due to restocking efforts from other European populations.

However, we need to realize some limitation about the study:

1. Sample Size and Representation: The study appears to have a limited sample size, especially concerning pre-1915 specimens. This could lead to a skewed representation of genetic diversity.

2. Temporal Resolution: The use of samples from two broad time periods (pre- and post-1915) might not capture finer temporal changes in the population's genetic structure.

3. Geographic Scope: While the study focuses on the Italian subspecies, it would be beneficial to include more comprehensive sampling from other European populations for comparison.

4. CR-Only Analysis: The exclusive focus on the mitochondrial control region may overlook nuclear DNA variations, which are crucial for a more comprehensive understanding of genetic diversity.

5. Potential for Contamination: Given the use of historical museum samples, there is a risk of contamination, which might not have been sufficiently addressed.

6. Lack of Functional Analysis: The study focuses on haplotype diversity without linking these findings to any functional implications or adaptive significance.

7. Conservation Implications: The paper suggests an uncertain future for the Italian subspecies but does not provide specific conservation recommendations based on the genetic data.

Therefore, some questions for the authors:

1. How was the sample size determined, and do you believe it is representative of the broader population?

2. Could you provide more details on the criteria used for selecting the time periods (pre- and post-1915)?

3. Were any nuclear DNA markers considered for inclusion in the study, and if not, why?

4. How did you address the potential for contamination in historical samples?

5. What steps were taken to validate the mitochondrial CR haplotypes identified in the study?

6. Can you expand on how restocking events were historically documented and correlated with genetic data?

7. What are the potential adaptive implications of the haplotype changes observed?

8. Did you find any evidence of selection acting on the mitochondrial control region in this population?

9. How do the findings of this study compare with similar studies on other European populations of P. perdix?

10. What conservation actions do you recommend based on the genetic findings of your study?

11. Were there any discrepancies in haplotype identification between different methodologies used in the study?

12. How might future research build on your findings to further inform conservation efforts?

We suggest a minor revision.

Author Response

RESPONSES TO REVIEWER 3

Thank you for your suggestions which we appreciated. Below we have inserted the responses to your observations and have tried to modify the text of the work according to your indications.

To facilitate reading the responses, we preferred to quote the text of the article when necessary to reference the part where the reviewer's request is addressed, rather than citing the line number, so that there is no need to go back into the text to find the indicated point.

Comment 1

Sample Size and Representation: The study appears to have a limited sample size, especially concerning pre-1915 specimens. This could lead to a skewed representation of genetic diversity.

Question 1

How was the sample size determined, and do you believe it is representative of the broader population?

Response to comment 1 and to question 1

It is indeed true that a small sample can lead to a underrepresentation of certain genotypes, but it is to take in account the nature of the analyzed samples. In fact, historic specimens dated before the beginning of the 20th century are not just hard to find but are also particularly difficult to process for molecular analyses. Nevertheless, we are confident that the museal sample of individuals collected before 1915 seams enough representative both geographically and genetically, since the analyses had to be carried out opportunistically on the only available specimens well preserved in museums. Note that we collected over 70 samples but only 40 preserved enough good quality DNA to be analyzed. The search for museum samples on a national scale was done in a very wide way, also involving the national association of museums and contacting facilities throughout Italy. In addition, we create a network with the University of Perugia and University of Piemonte Orientale in order to manage and share samples.

Therefore, we have changed the draft to the line 171 adding “The search for museum samples on a national scale was done in a very wide way, also involving the national association of museums and contacting facilities throughout Italy. In addition, we create a network with the University of Perugia and University of Piemonte Orientale in order to manage and share samples.”

And to the line 589 “Another possible limitation could be opportunistic sampling, based on the only available specimens well preserved in museums, that could lead to an underrepresentation of certain genotypes, but it is to take in account the nature of the analyzed samples. Historic specimens dated before the beginning of the 20th century are not just hard to find but are also particularly difficult to process for molecular analyses. “

Comment 2

Temporal Resolution: The use of samples from two broad time periods (pre- and post-1915) might not capture finer temporal changes in the population's genetic structure.

Question 2

Could you provide more details on the criteria used for selecting the time periods (pre- and post-1915)?

Response to comment 2 and to question 2

The partition of the samples in three broader time periods is justified by the locus that has been analyzed. In fact, the control region of the mitochondrial DNA, even if it’s a highly polymorphic non-coding locus, is characterized by slow mutation rates that do not make it an adequate tool to assess fine genetic changes that occur on short periods of time. Aside from this molecular reason, we decided to regroup the samples in clusters spaced by a few decades to avoid the creation of smaller and not statistically significant groups composed by few samples.

The division into the three groups was functional to the data obtained from the gray literature and therefore they were divided between the periods according to the numerical extent of the introductions and the number of specimens released, minimal and negligible before 1915, intermediate in the second period and extremely intense post World War II.

The original text (line 313) was changed as follow:

 “The analysis of the Historical information (grey literature) underlined a variation in both the frequence and consistency of the release of grey partridge in Italy. In particular, this practice was documented since the beginning of the 20th century (first range: before 1915). In this period animal releases were sporadic and numerically exiguous. Between 1915 and 1945, release events grew exponentially (hundreds of releases), with an understandable interruption during the Second World War. After 1945 the release practice showed a marked and continuous increment (thousands of releases). For these reasons, according to these data, the sampling was divided into three temporal datasets: before 1915 (n=54) considering very low level of allochthonous introgression; between 1915 and 1945 (n=29); and after 1946 (n=57) to verify any changes in the genetic composition. Furthermore, we chose to regroup the samples in clusters spaced by a few decades to avoid the creation of smaller and not statistically significant groups composed by few samples.”

Comment 3

Geographic Scope: While the study focuses on the Italian subspecies, it would be beneficial to include more comprehensive sampling from other European populations for comparison.

Response to comment 3

Museal samples from Northern Europe, as well as all the available sequences deposited in GenBank, and sequences retrieve from the polymorphisms detected by Liukkonen et al. 2002 were included in the median joining network. Furthermore, particular relevance was given to the Italian samples as the manuscript is focalized on the Italian subspecies and not on the species on a broader European context.

We add to the text at line 189: “To highlight the relationships between European contemporary, Italian museum and Liukkonen et al [24,32] haplotypes, 141 sequences were downloaded from GenBank, and used in statistical analyses (see table S3 on the supplementary materials).”

Comment 4

CR-Only Analysis: The exclusive focus on the mitochondrial control region may overlook nuclear DNA variations, which are crucial for a more comprehensive understanding of genetic diversity.

Question 3

Were any nuclear DNA markers considered for inclusion in the study, and if not, why?

Response to comment 4 and to question 3

The use of nDNA is certainly a great asset for phylogenetic analysis, especially to those carried out on finer scales. Unfortunately, the use of nDNA in museal samples is hindered by several factors, namely: the processing of the skin during tanning, the natural degradation of DNA and the fact that nDNA is present in a single copy inside cells. For these reasons mtDNA is much more indicated for phylogenetic studies on conservative and museal samples as it is more abundant inside cells, is protected by mitochondrial membranes and is comparatively shorter and less impacted by time-related degradation. The choice of analyzing the CR of the mtDNA also stemmed by the presence of abundant literature data to compare to the analyzed samples.

As it was usual for 18th century preparations, several specimens were mounted using arsenic as preservative (Marte et al. 2006) and administered several treatments of pesticides and freezing-thawing to get rid of potential pests that could have degraded the mount. All these circumstances are detrimental for the conservation of the genetic material and favor the fragmentation of the DNA that can then more easily be amplified choosing mitochondrial DNA and/or in shorter fragments (Straube et al. 2021; Sawyer et al. 2012).

Marte F., Péquignot A., Endt D. 2006. Arsenic in taxidermy collection: history, detection and management. Collection Forum 21(1).

Sawyer S., Krause J., Guschanski K., Savolainen V., Pääbo S. 2012. Temporal patterns of nucleotide misincorporations and DNA fragmentation in ancient DNA. PLoS One 7(3).

Straube N., Lyra M.L., Paijmans J.L.A., Preick M., Basler N., Penner J., Rödel, M.O., Westbury M.V., Haddad C.F.B., Barlow A., Hofreiter M. 2021. Successful application of ancient DNA extraction and library construction protocols to museum wet collection specimens. Molecular Ecology Resources 21: 2299– 2315

Comment 5

Potential for Contamination: Given the use of historical museum samples, there is a risk of contamination, which might not have been sufficiently addressed.

Question 4

How did you address the potential for contamination in historical samples?

Response to comment 5 and to question 4

To avoid contamination, the following precautions have been adopted as specified in the text

  • "All the protocols were performed under a hood after UV sterilization of plastics to reduce the risk of exogenous DNA contamination.
  • “A negative control was used to detect any DNA contamination event."

However, we did not specify that the different steps of the analyses (DNA extraction, pre-PCR, and post-PCR) are carried out in separate rooms to prevent amplified PCR products from contaminating the earlier stages and that most of the liquid handling steps are performed by robots under a hood. We also used positive as well as negative samples in each step and all the steps were carried out by working with a few samples at a time.

As proof that we have carefully considered the problem, 20 sequences showing signs of contamination, were eliminated from the analysis, as specified in the text: “20 presented evidence of contamination and 9 showed unreliable genetic results so they were removed from the following analysis.”

We have added the following part to the text (in red) at the line 220 “A negative and positive control was used to detect any DNA contamination event. All the different steps of the analyses (DNA extraction, pre-PCR, and post-PCR) are carried out in separate rooms to prevent amplified PCR products from contaminating the earlier stages and that most of the liquid handling steps are performed by robots under a hood. For the same purpose

all the steps were carried out by working with a few samples at a time.”

Question 5

What steps were taken to validate the mitochondrial CR haplotypes identified in the study?

Response to question 5

To validate the identified haplotypes "Sequences were deemed reliable only when sequences from both forward and reverse primers had been obtained" as specified in the text.

We also used specific software such as Seqscape v3.0 (Thermo Fisher Scientific) for the data correction phase. Seqscape is a resequencing package designed for mutation detection and analysis, SNP discovery and validation, allele identification, and sequence confirmation. It provides library functions for comparison to a known group of sequences

In cases of low sample yield, the DNA was re-extracted, followed by amplification and sequencing. Some of the sequences were amplified at different times, starting from the extraction, and the same results were obtained. For a sample of 21 sequences, we obtained 3 to 4 consistent results.

Comment 6

Lack of Functional Analysis: The study focuses on haplotype diversity without linking these findings to any functional implications or adaptive significance.

Question 7

What are the potential adaptive implications of the haplotype changes observed? 

Question 8

Did you find any evidence of selection acting on the mitochondrial control region in this population?

Response to Comment 6 and question 7 and 8

The focus of the article was on identifying historical haplotypes in museum samples of grey partridges present in Italy before the introduction of non-native specimens for hunting purposes, and the related differences compared to other European populations. The potential adaptive implications of these mutations were not investigated, especially considering that the mitochondrial DNA control region is non-coding and that our fragment is relatively short. However, it is worth noting that there are studies on farm chickens that link mutations in the mitochondrial DNA control region with growth rate (Xiujun T et al., 2022).

In the text, we conducted Tajima's and Fu and Li's tests to verify whether the dataset was under selection or neutral. The result is mostly not significant, except for Fu and Li's test, which is negative when examining the entire dataset. This is indicated in the text by the sentence: “After computing the Tajima’s test, the null hypothesis of neutral evolution was confirmed: Tajima’s D was negative in the subsample 1915-1945 and the whole sample dataset, even if not significantly, and close to zero in the other two groups. Values from Fu and Li's tests were not significant except for the whole dataset, which was significantly negative (Table 6).”

Xiujun T, Yanfeng F, Xiaoxu J, Qinglian G, Junxian L, Wei H, Honglin L, Yushi G. Haplotype study of the mitochondrial control region of broiler breeds with different growth rates. Anim Biotechnol. 2023 Dec;34(7):3165-3173. doi: 10.1080/10495398.2022.2138412. Epub 2022 Oct 30. PMID: 36309842.

Comment 7

Conservation Implications: The paper suggests an uncertain future for the Italian subspecies but does not provide specific conservation recommendations based on the genetic data.

Question 10

What conservation actions do you recommend based on the genetic findings of your study?

Response to Comment 7 and to question 10

This issue was partially already analysed in the discussion section (line 642) where was the sentence "The molecular characterizations of historical Italian populations from up to 1999 and the comparison and network with European data represent a fundamental step not just for future studies, but also to direct breeding plans to carry out restocking interventions with native genotypes in the framework of management projects such as Life Perdix - Life17 NAT/IT/000588" that was rephrased as follow:

“The molecular characterization of historical Italian populations from up to 1999 and the comparison and network with European data represent a key-step for conservation purposes. In fact, for first these data will be a fundamental reference for future studies, based either on historical or actual specimens. Furthermore, these data will also drive direct breeding plans to carry out restocking interventions with native genotypes in the framework of management projects. The Life Perdix - Life17 NAT/IT/000588 is a project representing an example of how information derived from historical data could be used and applied for conservation and management purposes. The preservation of DNA from museum specimens does not prevent the disappearance of specific genotypes, but it can compensate for the great loss of knowledge that accompanies the extinction of animal species. In particular, the genotyping of specimens from museum collections does not only represent a cognitive data in itself but provides indications on the modern genotypes most similar to the historical ones to be used for any restocking actions. In particular, in the case of the grey partridge present in Italy, we highlighted that it exhibited unique haplotypes not found in European populations. Therefore, in the event of reintroductions, it will be important to consider these factors in addition to ensuring the necessary genetic variability.”

Question 6

Can you expand on how restocking events were historically documented and correlated with genetic data?

Response to question 6

We have rewritten the section related to historical documentation, as explained in the response to comment 2, to better clarify how the historical stocking events were documented. The correlation with the genetic data is descriptive and observational. We know that in the three historical frames, there was an exponential increase in releases, and we observe genetic data indicating that some genotypes present before the massive introductions have disappeared, while others have appeared such as haplotype Pdx_E1 typical of Eastern European populations, during the periods of the most documented introductions.

We have integrated the text of the article in the line 607 as follows: “The historical data obtained from the review of grey literature helps us determine up to what period we can consider the haplotypes as representative of the Italian population. It also confirms that the dates identified as the beginning of the massive introduction of non-native individuals coincide with the discovery of haplotype Pdx_E1 in an animal compatible with those dates.”

Question 9

How do the findings of this study compare with similar studies on other European populations of P. perdix?

Response to question 9

The paper reported the genetic characterization of Italian historical specimens, comparing obtained ancient haplotypes for other European countries. The already reported comparison (figure 4) contextualize the historical haplotypes in the European genetic pool. Considering that this study aims to characterize the genotypes shown by Italian museum specimens, with particular relevance to the Italian subspecies, further analyses involving northern European datasets wouldn't have a relevant informative value for the paper's purposes.

There are few studies on this topic, and it is often difficult to compare data due to factors such as the use of different DNA fragments. However, we did compare our data with the work of Liukkonen et Al. and Andersen et Al, as indicated in the paper in line 552, as add at line 189 and 455.

“The high value of haplotype diversity and nucleotide diversity that was found in each time-range group, comparing other similar studies (e.g. Andersen et al., 2011 – [34]) could be due to the repeated and continuous restocking actions using individuals from different regions and other subspecies.”

“To highlight the relationships between European contemporary, Italian museum and Liukkonen et al [24,32] haplotypes, 141 sequences were downloaded from GenBank, and used in statistical analyses (see table S3 on the supplementary materials).”

“Note that W2 and W28 haplotypes in the network correspond to a unique circle but considering full length in Liukkonen et al. [24] sequences are separated by four mutations.”

“On the contrary, the metapopulation seems subdivided at geographical scale, as already reported following the distinction among PW and PE haplotypes [24].”

Question 11

Were there any discrepancies in haplotype identification between different methodologies used in the study?

Response to question 11

A unique genetic method was used, using 2 types of primer to optimize amplification. One pair amplify a longer fragment and the other an internal smaller one. The other method is related to bibliographic research on the literature of the time but does not help us identify the haplotypes, only the time frames.

Question 12

How might future research build on your findings to further inform conservation efforts?

Response to question 12

We believe that our research can establish a Baseline for the Italian grey partridge and be helpful in comparing existing data in GenBank by providing a unique identifier for all deposited haplotypes. We have also laid the groundwork for future reintroductions in Italy. Thanks to the grey literature (often overlooked) that we thoroughly examined for the first time, we were able to identify the dates of introductions carried out in Italy.

we added to the text of the work to the line 665 and 642 “Another crucial objective of this work, in addition to identifying and publishing the haplotypes exclusive of the Italian territory before the massive introductions, was to clarify and unify all the haplotypes publicly available in GenBank by assigning them a unique identifier, with the aim of laying the basis for future research studies.”….

…..”The molecular characterization of historical Italian populations from up to 1999 and the comparison and network with European data represent a key-step for conservation purposes. In fact, for first these data will be a fundamental reference for future studies, based either on historical or actual specimens.”
